# Mitigating Reward Extrapolation Errors in Offline Preference-Based RL via Attention-Guided Subgoal Discovery

## Abstract

Offline preference-based reinforcement learning (PbRL) learns complex behaviors from human feedback without environment interaction, but suffers from reward model extrapolation errors when encountering out-of-distribution region during policy optimization. These errors arise from distributional shifts between preference-labeled training trajectories and unlabeled inference data, leading to reward misestimation and suboptimal policies. We introduce SPOT (Subgoal-based Preference Optimization Through Attention Weight), which mitigates extrapolation errors by leveraging attention-derived subgoals from preference data. SPOT regularizes the policy toward subgoals observed in preferred trajectories. This approach constrains learning within the training distribution, reducing reward model extrapolation errors. Through comprehensive experiments, we demonstrate that our subgoal-guided approach achieves superior performance compared to existing methods while reducing extrapolation errors. Our approach preserves fine-grained credit assignment information while enhancing query efficiency, suggesting promising directions for reliable and practical offline preference-based learning.

## 1 Introduction

Preference-based reinforcement learning (PbRL) has demonstrated remarkable success across diverse domains. PbRL learns reward functions directly from human feedback, eliminating the overhead of manually designing the dense reward functions (Christiano et al., 2017b). This paradigm is particularly valuable in complex scenarios where defining precise reward functions is challenging, such as robotic manipulation (Akrour et al., 2011), autonomous driving (Surmann et al., 2025), and LLMs (Fernandes et al., 2023; Korbak et al., 2023). With the growing utilization of offline data in policy optimization (Fang et al., 2022; Prudencio et al., 2023), offline PbRL has emerged as a significant area of research (Tu et al., 2025).

The standard offline PbRL framework follows a two-stage process. First, a reward model is trained using pairwise preference-labeled trajectory datasets to approximate step-wise rewards. Second, this learned reward model is used to label an unlabeled trajectory dataset, which is then utilized for policy optimization through reinforcement learning algorithms. Offline PbRL faces fundamental challenge in learning accurate step-wise reward model from coarse-grained trajectory-level preferences. This challenge stems primarily from extrapolation errors –a critical limitation when reward models encounter distributional shifts (Yu et al., 2022; Gulcehre et al., 2021). Specifically, trajectories used for policy optimization often lie outside the distribution of preference-labeled data, creating out-of-distribution regions where reward model estimates become unreliable. These estimation errors can significantly mislead policy learning by providing over- or underestimated reward signals, which in turn leads suboptimal performance by either inflated Q-function estimates or deflated value estimates (Fujimoto et al., 2019; Kumar et al., 2020).

Two main directions were suggested to mitigate this challenge: improving reward model reliability (Tu et al., 2025) or completely eliminating them (Hejna & Sadigh, 2023; An et al., 2023). While these approaches do reduce reward model extrapolation errors, they overlook the rich information

contained in preference datasets, dismissing valuable signals that could further alleviate extrapolation error.

Building on recent advances in attention-based reward modeling (Kim et al., 2023; Verma & Susa, 2024), we observe that preference-based RL identifies critical states within trajectories through attention mechanisms that assign higher weights to states strongly influencing human preferences. We conceptualize these high-attention states as **subgoals**, which act as critical decision points or milestones. These subgoals are anchored within the demonstrated preferred trajectories, which helps to mitigate extrapolation errors while simultaneously providing auxiliary waypoints that guide reward learning through additional supervisory structure.

In this work, we propose **SPOT** (Subgoal-based Preference Optimization Through Attention Weight), a novel approach that addresses reward model extrapolation errors in offline PbRL. Our approach improves reward model reliability by utilizing meaningful subgoals extracted from high-attention weight points on preferred trajectories. We employ a Conditional Variational Autoencoder (CVAE) to learn the underlying distribution of these preference-aligned subgoals, enabling generation of contextually appropriate intermediate subgoals for unlabeled trajectories. By incorporating subgoals as intermediate reward signals, SPOT effectively mitigates extrapolation errors while preserving fine-grained credit assignment information. SPOT regularizes the policy toward subgoals observed in preferred trajectories. Through empirical evaluation, we demonstrates that SPOT achieves state-of-the-art performance across multiple benchmarks while effectively addressing extrapolation errors and improving reward model reliability.

## 2 RELATED WORK

Offline Preference-based Reinforcement Learning (PbRL) has emerged as a promising paradigm that combines human preference feedback with offline RL to learn effective policies without online environment interaction (Christiano et al., 2017b; Lee et al., 2021; Liang et al., 2022; Park et al., 2022). The traditional approach follows a two-stage framework: first learning a reward function from human preference data, then applying standard reinforcement learning algorithms (Haarnoja et al., 2018; Schulman et al., 2017) using the learned reward function for policy optimization. Recent advances have enhanced offline preference learning through non-Markovian reward structures (Kim et al., 2023), contrastive learning frameworks (Hejna et al.), and data augmentation techniques (Choi et al., 2025). Modern approaches integrate diffusion models for trajectory optimization (Zhang et al., 2024) and leverage large language models for preference elicitation (Ouyang et al., 2022; Verma & Susa, 2024; Early et al., 2022; Kang et al., 2023).

Existing offline RL suffers from extrapolation error due to distribution mismatch, leading to either overestimated Q-values for out-of-distribution actions (Gulcehre et al., 2021) or deflated value estimates (Yeom et al., 2024). Various error regularization methods address this challenge, including BCQ (Fujimoto et al., 2019), CQL (Kumar et al., 2020), and IQL (Kostrikov et al., 2021), which constrain learning OOD region. Reward shaping provides another principled approach to address extrapolation error with policy invariance guarantees (Ng et al., 1999). Techniques include positive reward shaping for offline dataset conservative exploitation (Sun et al., 2022), adaptive shaping mechanisms (Zhang & Tan, 2023; Rezaeifar et al., 2022), and model-based penalties (Yu et al., 2020). Recent work extends this through language-guided (Goyal et al., 2019) and goal-conditioned formulations (Mezghani et al., 2022). In offline PbRL, extrapolation errors are further amplified than in offline RL due to the existence of the reward model. Distribution mismatch between preference-labeled trajectories and policy optimization trajectories causes biased reward estimates (Yu et al., 2022; Konyushkova et al., 2020; Hu et al., 2023). Recent approaches address this through trajectory return regularization (Tu et al., 2025) or alternative paradigms that circumvent explicit reward modeling (Hejna & Sadigh, 2023; An et al., 2023) by directly optimizing against preference datasets.

## 3 PRELIMINARIES

**Offline Preference based Reinforcement Learning** Traditional offline PbRL approaches employ a Markov Decision Process (MDP) (Christiano et al., 2017a) framework for preference learning. Let $\sigma^{(\ell)} = (s_1^{(\ell)}, a_1^{(\ell)}), \ldots, (s_H^{(\ell)}, a_H^{(\ell)})$, where $\ell \in 0, 1$. preferences are collected as triples $(\sigma^0, \sigma^1, y)$,

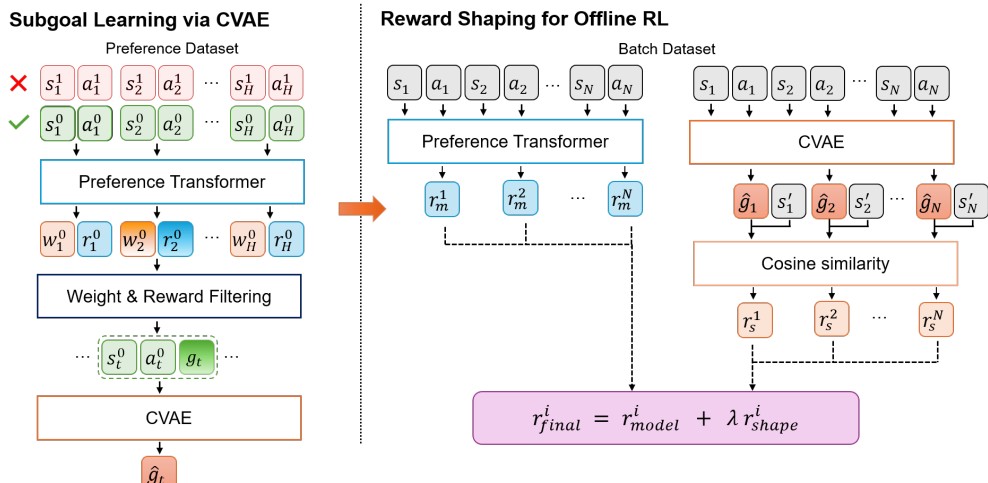

Figure 1: Overall architecture of *SPOT*. Our framework consists of two main stages: (1) Subgoal Learning via CVAE (left): The Preference Transformer, as a reward model, processes state-action pairs $(s_t, a_t)$ and produces attention weights $w_t$ and rewards $r_t$ during reward learning. Subgoal states $S_g$ are identified by applying weight and reward filtering, selecting states with both top K% attention weights and above-average reward values. The CVAE learns to generate subgoal $\hat{g}$ conditioned on each intermediate state and action. (2) Reward Shaping for offline RL (right): For training, the batch dataset is simultaneously processed through both the Preference Transformer to obtain model rewards $r_m$ and the CVAE to generate predicted subgoals $\hat{g}$. The final reward $r_{final}$ is computed by combining the model reward with a shaped reward term derived from cosine similarity between predicted subgoals and next states, weighted by hyperparameter $\lambda$.

where $y \in \{0, 1, 0.5\}$ denotes the preference label: $y = 1$ if $\sigma^1 \succ \sigma^0$, $y = 0$ if $\sigma^0 \succ \sigma^1$, and $y = 0.5$ for equal preference. The Bradley-Terry model (Bradley & Terry, 1952) with Markovian reward assumption is typically employed (Christiano et al., 2017a) :

$$P[\sigma^1 \succ \sigma^0; \psi] = \frac{\exp(\sum_t r_\psi(\mathbf{s}_t^1, \mathbf{a}_t^1))}{\sum_{j \in \{0,1\}} \exp(\sum_t r_\psi(\mathbf{s}_t^j, \mathbf{a}_t^j))} \tag{1}$$

This approaches are trained using cross-entropy loss with human-provided preference labels $y$:

$$\mathcal{L}_{CE} = -\mathbb{E}_{(\sigma^0, \sigma^1, y) \sim \mathcal{D}} \left[ y \log P[\sigma^1 \succ \sigma^0; \psi] + (1 - y) \log P[\sigma^0 \succ \sigma^1; \psi] \right] \tag{2}$$

**Preference Transformer** Preference Transformer (PT) (Kim et al., 2023) formulates preference learning as a non-Markovian reward problem (Bacchus et al., 1996). PT employs a causal transformer to process state-action sequences and a preference attention layer to generate non-Markovian rewards $\hat{r}$ and importance weights $w_t$. Each trajectory segment is processed through a causal transformer backbone, followed by a bidirectional attention mechanism that produces both predicted scalar rewards and associated attention weights at each timestep. The preference prediction is formulated as:

$$P[\sigma^1 \succ \sigma^0; \psi] = \frac{\exp\left(\sum_t w((s_i^1, a_i^1)_{i=1}^H; \psi) \cdot \hat{r}((s_i^1, a_i^1)_{i=1}^t; \psi)\right)}{\sum_{l \in 0,1} \exp\left(\sum_t w((s_i^l, a_i^l)_{i=1}^H; \psi) \cdot \hat{r}((s_i^l, a_i^l)_{i=1}^t; \psi)\right)} \tag{3}$$

where the reward function $\hat{r}_\psi$ takes into account the trajectory history $\{(s_i, a_i)\}_{i=1}^t$ and the attention weights $w$ are computed over the previous $H$ steps. This approach enables credit assignment through importance weights $w_t^i$.

# 4 METHOD

We propose an enhanced offline PbRL framework that integrates attention-driven subgoal discovery to mitigate extrapolation error. Our approach extends the traditional two-phase PbRL paradigm by incorporating a novel subgoal learning mechanism in the first phase and leveraging these learned subgoals for effective reward shaping in the second phase. Our framework addresses the extrapolation error problem by constraining policy learning toward subgoals where reward models produce unreliable estimates. The framework simultaneously trains a CVAE during reward model learning and applies the learned subgoal guidance during offline RL training.

## 4.1 SUBGOAL LEARNING VIA CVAE

### 4.1.1 ATTENTION-BASED SUBGOAL IDENTIFICATION

Building upon the Preference Transformer architecture Kim et al. (2023), which employs causal transformers with bidirectional attention layers for credit assignment in preference trajectories, we leverage attention weights as importance measures to identify critical states within trajectories. The attention mechanism captures states that most strongly influence human preferences. This attention weight can capture the temporal dependencies and state importance that are crucial for subgoal identification.

For a given trajectory segment $\sigma = \{(s_t, a_t)\}_{t=1}^H$, we extract attention weights $w_t$ through the preference transformer:

$$w_t = f_{\text{attention}}(s_t, a_t; \theta) \tag{4}$$

where $f_{\text{attention}}$ represents the attention mechanism parameterized by $\theta$, producing scalar attention weights that quantify the importance of each state-action pair in the trajectory.

### 4.1.2 DUAL-CRITERIA FILTERING

In preferred trajectories that only marginally outperform non-preferred ones, high attention states are prone to focus on relatively bad states. To avoid selecting less desirable subgoals, we introduce a dual-criteria filtering mechanism, attention-based and reward-based criteria. The subgoal state set $\mathcal{S}_g$ is then constructed by selecting states that satisfy both criterias:

$$\mathcal{S}_g(\sigma; K) = \{s_t \mid w_t \geq \alpha_K(\sigma) \wedge \hat{r}_t \geq \bar{r}(\sigma)\} \tag{5}$$

$$\alpha_K(\sigma) = \text{Quantile}_{1-K\%}\left(\{w_i\}_{i=1}^T\right) \tag{6}$$

where $\alpha_K(\sigma)$ represents the $(100-K)$-th percentile threshold of attention weights within trajectory $\sigma$, ensuring we select only the top $K\%$ attention states. The reward constraint $\hat{r}_t \geq \bar{r}(\sigma)$ with $\bar{r}(\sigma) = \frac{1}{T}\sum_{i=1}^T \hat{r}_i$ selects states that exceed the trajectory's average reward. This dual-criteria approach serves a critical role in extrapolation error mitigation by guaranteeing that high-quality subgoals are derived exclusively from preference-aligned training trajectory segments.

### 4.1.3 CONDITIONAL VARIATIONAL AUTOENCODER TRAINING

Although our method identifies meaningful subgoals in preferred trajectories, applying them to unlabeled data presents a key challenge: mapping these waypoints to arbitrary state-action pairs during policy optimization. To address this, we employ a Conditional Variational Autoencoder (CVAE) that learns the underlying distribution of preference-aligned subgoals and generates contextually relevant subgoals conditioned on current state-action. This enables SPOT to provide appropriate intermediate guidance during policy optimization.

CVAE is trained with state-action-subgoal triplets $(s_t, a_t, g_t)$ sampled from preferred trajectories, where $s_t$ and $a_t$ is a corresponding state-action pairs between $g_{t-1}$ and $g_t$. The CVAE framework models the conditional distribution $p_\theta(g|s_t, a_t)$ through three components:

- **Encoder network**: $q_\phi(z|g_t, s_t, a_t)$ that approximates the posterior distribution
- **Prior network**: $p_\psi(z|s_t, a_t)$ that models the latent space conditioned on current context
- **Decoder network**: $p_\theta(g_t|z, s_t, a_t)$ that reconstructs subgoals from latent representations

The CVAE training objective combines reconstruction accuracy with regularization:

$$\mathcal{L}_{\text{CVAE}} = -\mathbb{E}_{q_\phi(z|g_t,s_t,a_t)} \left[\log p_\theta(g_t|z,s_t,a_t)\right] + \beta D_{\text{KL}} \left(q_\phi(z|g_t,s_t,a_t)\|p_\psi(z|s_t,a_t)\right) \quad (7)$$

To maintain directional consistency between current states and target subgoals, we introduce an additional cosine similarity loss:

$$\mathcal{L}_{\text{sim}} = -\frac{1}{2} \left(1 + \frac{\hat{g}_t \cdot g_t}{\|\hat{g}_t\|\|g_t\|}\right) \quad (8)$$

where $\hat{g}_t$ represents the CVAE-generated subgoal and $g_t$ is the ground-truth subgoal. The complete training objective is:

$$\mathcal{L}_{\text{total}} = \mathcal{L}_{\text{CVAE}} + \mathcal{L}_{\text{sim}} \quad (9)$$

The CVAE framework ensures that generated subgoals remain within the training distribution. This is achieved via the KL divergence term in the objective function, which regularizes the latent space to prevent the decoder from generating out-of-distribution subgoals.

## 4.2 Reward Shaping for Offline RL

### 4.2.1 Sub-goal-Guided Reward Augmentation

The learned CVAE generates contextually relevant sub-goals during offline RL training. For each state-action pair $(s_i, a_i)$ in a training batch $\mathcal{B} = \{(s_i, a_i)\}_{i=1}^N$, we generate corresponding subgoals:

$$\hat{g}_i = G_\phi(s_i, a_i), \quad \forall(s_i, a_i) \in \mathcal{B} \quad (10)$$

where $G_\phi$ represents the trained CVAE decoder network.

To measure progress toward these generated sub-goals, we compute a normalized similarity between the next state $s_i'$ and the predicted sub-goal $\hat{g}_i$:

$$\text{sim}(s_i', \hat{g}_i) = \frac{s_i' \cdot \hat{g}_i}{\|s_i'\|\|\hat{g}_i\|} \quad (11)$$

$$r_{\text{shape}}(s_i', \hat{g}_i) = \frac{\text{sim}(s_i', \hat{g}_i) + 1}{2} \quad (12)$$

The normalization ensures $r_{\text{shape}} \in [0, 1]$, providing a consistent scale for reward combination. The resulting similarity-based reward provides an auxiliary signal that guides the policy toward preference-aligned subgoals. This mechanism effectively constrains the policy to regions well-supported by the training data and thereby mitigating catastrophic extrapolation errors.

### 4.2.2 Integrated Reward Signal

The final reward is the weighted sum of the original reward model output and the subgoal-based shaping term:

$$r_{\text{final}}(s_i, a_i, s_i') = r_{\text{model}}(s_i, a_i) + \lambda r_{\text{shape}}(s_i', \hat{g}_i) \quad (13)$$

where $\lambda \in [-1, 1]$ is a carefully chosen hyperparameter that balances the contribution of subgoal guidance without overwhelming the primary reward signal. This formulation preserves the original task objectives while providing auxiliary guidance toward meaningful intermediate states.

## 5 Experiment

**Benchmarks** We evaluate our approach against state-of-the-art offline preference-based RL methods on three widely-adopted benchmarks: D4RL Gym Locomotion (Fu et al., 2020), Robosuite robomimic (Mandlekar et al., 2021), and Meta-World (Yu et al., 2019). Following established protocols from prior work (Brockman et al., 2016; Zhu et al., 2025), we conduct evaluations across diverse task domains and report average normalized scores for D4RL and success rates for Robomimic and Meta-World.

Table 1: Performance Comparison Across Models and Tasks. We report average normalized scores on Gym-MuJoCo locomotion tasks in D4RL and success rates on Robosuite and Meta-World manipulation tasks. For D4RL tasks, *hop* and *walk* represent hopper and walker2d, where *m*, *r*, and *e* denote medium, replay, and expert, respectively. For Robosuite tasks (lift, can), *ph* and *mh* denote proficient-human and multi-human datasets. For Meta-World, we evaluate on drawer-open and plate-slide tasks. All scores are reported as mean ± std across 5 random seeds, with **bold** indicating methods within the top 95% performance. Average performance across all tasks is shown in the final column. Note that oracle average is computed over 8 tasks excluding Meta-World.

| Dataset | Oracle | MR | PT | IPL | HPL | CPL | DTR | SPOT(ours) |
|---|---|---|---|---|---|---|---|---|
| *D4RL Locomotion Tasks* | | | | | | | | |
| hop-m-r | **92.02 ± 7.23** | 37.21 ± 12.53 | 52.15 ± 25.94 | 74.96 ± 5.79 | 79.89 ± 10.01 | 62.21 ± 6.40 | **94.18 ± 0.28** | 85.08 ± 1.32 |
| hop-m-e | 62.10 ± 30.42 | 63.60 ± 25.42 | 74.46 ± 4.33 | 42.11 ± 8.93 | 95.30 ± 10.66 | 44.97 ± 44.74 | **102.12 ± 6.79** | 98.73 ± 7.50 |
| walk-m-r | 67.59 ± 7.91 | 71.39 ± 2.66 | **73.85 ± 3.18** | 47.05 ± 15.24 | 49.89 ± 10.49 | 36.10 ± 12.61 | 69.09 ± 4.85 | 76.89 ± 2.46 |
| walk-m-e | **108.72 ± 1.86** | **110.88 ± 0.76** | 110.6 ± 0.43 | 107.78 ± 0.95 | 103.14 ± 2.49 | **108.98 ± 0.15** | 110.96 ± 0.37 | 110.06 ± 0.28 |
| *Robosuite Manipulation Tasks* | | | | | | | | |
| lift-mh | 81.62 ± 5.54 | **95.62 ± 2.23** | 68.46 ± 10.02 | 84.49 ± 4.28 | 88.37 ± 3.06 | 18.79 ± 5.19 | 22.30 ± 21.96 | 65.17 ± 12.57 |
| lift-ph | **98.43 ± 1.15** | 87.40 ± 10.65 | **95.50 ± 1.90** | **95.81 ± 3.04** | 61.04 ± 7.61 | 28.41 ± 5.85 | 9.86 ± 4.31 | **97.12 ± 1.81** |
| can-mh | 34.30 ± 6.95 | 47.95 ± 2.29 | 53.06 ± 14.48 | 41.12 ± 2.21 | 35.19 ± 12.25 | 12.34 ± 5.44 | **60.28 ± 2.56** | **60.55 ± 1.65** |
| can-ph | **73.25 ± 2.70** | 51.90 ± 6.58 | 48.74 ± 5.82 | 67.98 ± 3.41 | 10.90 ± 4.33 | 9.15 ± 2.40 | 39.82 ± 8.25 | 63.82 ± 5.64 |
| *Meta-World Manipulation Tasks* | | | | | | | | |
| drawer-open | — | **86.6 ± 14.3** | 42.8 ± 29.1 | **87.64 ± 6.99** | 83.13 ± 12.64 | 75.48 ± 7.42 | 26.90 ± 24.09 | 66.80 ± 18.05 |
| plate-slide | — | 51.5 ± 11.9 | 51.0 ± 2.8 | 51.18 ± 6.63 | 28.73 ± 12.22 | 53.41 ± 4.94 | 5.24 ± 5.07 | **64.0 ± 4.1** |
| **Average** | 77.25 | 73.61 | 74.76 | 73.24 | 67.96 | 44.98 | 54.08 | **78.82** |
| **Avg. Std** | 11.89 | 11.51 | 13.80 | **6.95** | 9.36 | 9.51 | 7.85 | 7.76 |

**Baselines** For comparative analysis, we establish a comprehensive set of baselines encompassing several key approaches in preference-based learning. These include Oracle reward (ground-truth reward from the dataset), Markovian Reward (MR) (Christiano et al., 2017a), Preference Transformer (PT) (Kim et al., 2023), Inverse Preference Learning (IPL) (Hejna & Sadigh, 2023), Hindsight Preference Learning (HPL) (Gao et al., 2024), Contrastive Preference Learning (CPL) (Hejna et al.), and In-Dataset Trajectory Return Regularization (DTR) (Tu et al., 2025). We adopt Implicit Q-Learning (IQL) (Kostrikov et al., 2021) as our core reinforcement learning algorithm, given its established track record in previous research. Each baseline method offers distinct characteristics: MR employs the Bradley–Terry model for preference-based reward extraction, PT implements a causal transformer architecture for non-Markovian reward inference, IPL demonstrates reward-free preference learning, HPL utilizes a variational autoencoder framework to predict future segments for reward labeling, CPL combines a regret-based preference model with contrastive objectives over preferred and non-preferred trajectory segments, and DTR regularizes policy learning toward high-return in-dataset trajectories to stabilize offline RL under learned rewards.

**Setup** The experimental setup utilize a training configuration wherein the importance weight Top-K% is set to 10, KL divergence term $\beta$ is fixed to 1, and the reward coefficient $\lambda$ is fixed at 1. Ablation studies about Top-K% at Section 5.2.1.

## 5.1 BENCHMARK RESULT

Our empirical results demonstrate that reward shaping with predicted subgoals significantly enhances the performance of offline Preference-based RL. Table 1 presents a comprehensive evaluation, confirming the consistent superiority of our approach across multiple benchmarks. In the hopper environment, SPOT achieves state-of-the-art performance on both medium-replay and medium-expert datasets, significantly outperforming existing benchmarks while maintaining notably low variance. The walker2d environment further validates our method's effectiveness, exhibiting remarkable stability across various data distributions. In manipulation tasks, our approach demonstrates consistent efficacy across different levels of demonstration quality, consistently achieving or approaching top-tier performance metrics. For meta-world, our method yields modest but meaning-

ful improvements over baseline approaches. Particularly noteworthy is the substantial performance enhancement in the drawer-open task compared to PT, despite its historically challenging low-reward characteristics, though it falls short of the absolute peak performance while still maintaining incremental improvements. Importantly, our approach achieves the highest mean performance of 78.82 across all evaluated tasks, substantiating the effectiveness of incorporating attention-guided subgoals in the offline preference-based reinforcement learning paradigm. Additionally, it demonstrates significantly reduced average standard deviation from 13.80 (PT) to 7.76.

## 5.2 ABLATION STUDY

### 5.2.1 ANALYSIS OF TOP-K% SUBGOAL PERFORMANCE

Table 2: Performance analysis across different Top-K% percentile groups.

| Percentile | hopper-medium-expert | Can-mh |
|---|---|---|
| Top 10% (SPOT) | **99.37 $\pm$ 8.35** | **59.56 $\pm$ 0.23** |
| Top 10–20% | 83.19 $\pm$ 2.85 | 54.10 $\pm$ 7.38 |
| Bottom 10–20% | 69.90 $\pm$ 39.12 | 50.38 $\pm$ 12.79 |
| Bottom 10% | 55.24 $\pm$ 24.39 | 50.04 $\pm$ 3.67 |

The analysis of performance across different Top-K% groups over 3 seeds reveals interesting patterns in how the importance weights correlate with performance. In both the hopper-medium-expert-v2 and Can-mh environments, we observe a clear hierarchical performance pattern that aligns with the percentile rankings. The top 10% group achieves the highest performance, followed closely by the top 10-20% group. This suggests that the higher importance weights effectively identify critical subgoal within the trajectories. Notably, there is a substantial performance gap between the upper and lower percentile groups in both environments. The bottom 10-20% group shows a second loweset performance with significantly higher variance in performance, while the bottom 10% group exhibits the lowest performance compared to other percentile groups. This increasing variance in lower percentiles suggests that lower attention weight subgoals may lead to more unstable performance outcomes. These findings suggest that the strategic extraction of subgoals significantly enhances reinforcement learning outcomes through more effective reward shaping mechanisms.

### 5.2.2 ANALYSIS OF REWARD SHAPING METHODS AND WEIGHT SELECTION

We conduct comparative analysis on different weight magnitudes ($\lambda \in [-1, 1]$) over three widely-used reward shaping methods.

1. **Negative Distance**: the Euclidean distance between current states and predicted subgoals.
2. **Potential-based** (Ng et al., 1999): Traditionally guaranteeing policy invariance with ground-truth rewards where policy invariance cannot be ensured with predicted rewards
3. **Cosine Similarity**: Capturing semantic relationships between states and predicted subgoals

Table 3 demonstrates that cosine similarity achieves superior performance on both environments. The potential-based method shows good performance on walker but higher variance on hopper, while negative distance exhibits sensitivity to weight selection with instability on walker. Weight analysis reveals that positive weights generally yield more stable performance, with weight 1.0 being particularly effective for cosine similarity. This indicates that positive reinforcement toward subgoals outperforms penalizing deviation, and that semantic relationships provide more informative guidance than other reward shpaing methods for policy learning.

## 5.3 EXTRAPOLATION ERROR ANALYSIS IN SPOT

To validate SPOT's effectiveness at mitigating extrapolation error, we analyze how proximity to predicted subgoals influences extrapolation errors. We define extrapolation error as the absolute difference between predicted reward and ground truth reward. Since true ground-truth rewards are unavailable in real environments, we use human-labeled rewards from the dataset as proxy ground

Table 3: Performance (mean ± std) of reward shaping on hopper-medium-expert and walker2d-medium-replay, averaged over 3 seeds.

| Env | Method | Weight ($\lambda$) | | | | | |
|-----|--------|------|------|------|------|------|------|
| | | $-1.0$ | $-0.5$ | $-0.1$ | $0.1$ | $0.5$ | $\mathbf{1.0}$ |
| **hop-m-e** | negative distance | $43.09 \pm 40.01$ | $64.32 \pm 44.12$ | $75.12 \pm 30.83$ | $49.27 \pm 41.61$ | $55.01 \pm 27.28$ | $86.03 \pm 9.77$ |
| | potential based | $51.01 \pm 45.45$ | $62.54 \pm 41.23$ | $96.03 \pm 3.14$ | $84.98 \pm 11.87$ | $45.80 \pm 49.24$ | $77.95 \pm 36.02$ |
| | **cosine similarity** | $62.78 \pm 38.47$ | $44.28 \pm 46.02$ | $56.65 \pm 33.46$ | $55.85 \pm 42.94$ | $63.89 \pm 51.95$ | $\mathbf{97.36 \pm 10.26}$ |
| **walk-m-r** | negative distance | $19.38 \pm 6.41$ | $13.93 \pm 1.18$ | $49.80 \pm 19.67$ | $71.23 \pm 2.38$ | $0.09 \pm 0.62$ | $0.23 \pm 0.06$ |
| | potential based | $75.47 \pm 2.20$ | $76.71 \pm 1.53$ | $76.15 \pm 3.72$ | $75.26 \pm 0.98$ | $74.45 \pm 5.41$ | $50.60 \pm 17.71$ |
| | **cosine similarity** | $0.69 \pm 1.60$ | $75.83 \pm 1.39$ | $74.84 \pm 0.78$ | $76.66 \pm 1.96$ | $75.30 \pm 2.73$ | $\mathbf{77.51 \pm 2.60}$ |

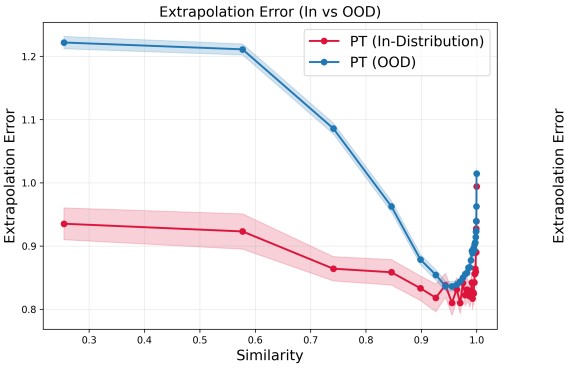 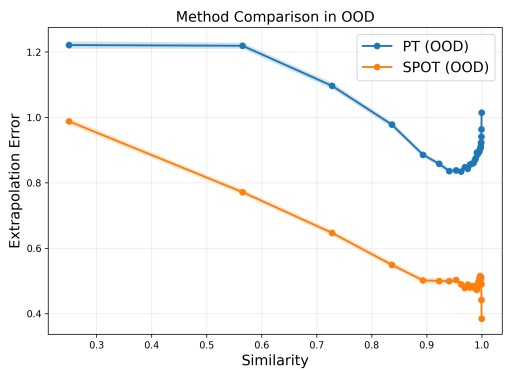

(a) Extrapolation Error: In-Distribution vs OOD.          (b) Extrapolation Error: PT vs. SPOT (OOD)

Figure 2: Extrapolation error analysis based on proximity to predicted subgoals. where a higher similarity value indicates closer proximity. (a) Extrapolation error of the PT on in-distribution versus out-of-distribution (OOD) data. (b) A direct comparison of extrapolation error between PT and our method, SPOT, in OOD setting.

truth. We measure distributional proximity using cosine similarity between the predicted subgoal state and the current state. In figure 2a, We evaluate the performance under two distributional settings: in-distribution setting only on the reward model training data, and out-of-distribution (OOD) setting on trajectories used during policy optimization that exclude from training data. The result confirms that out-of-distribution (OOD) scenarios exhibit substantially higher prediction errors compared to in-distribution data. States with high similarity to subgoals tend to exhibit reduced extrapolation errors. Figure 2b demonstrates that as cosine similarity approaches 1, the extrapolation error significantly reduces for both methods. Notably, SPOT consistently outperforms the Preference Transformer (PT) baseline, showing substantially lower extrapolation errors across all distance ranges. Subgoal-guided reward shaping approach effectively reduces this extrapolation gap particularly in OOD settings compared to PT, demonstrating its robustness in handling distribution shifts through structured intermediate goal prediction.

## 5.4 SUBGOAL EXTRACTION CASE STUDY

Figure 3 demonstrates the forward-looking nature of our subgoal extraction mechanism through a qualitative analysis in the hopper environment. We compare the original observations with their corresponding predicted subgoals during critical phases of a jumping. During the pre-jump phase (Figure 3a), the agent exhibits a preparatory stance, while the predicted subgoal (Figure 3b) shows an optimal jumping with extended limbs and forward momentum. Conversely, during the jumping phase (Figure 3c), when the agent is mid-air, the corresponding subgoal (Figure 3d) proactively displays a landing-ready posture with bent joints positioned for safe ground contact. Our case study clearly shows that critical moments captured via subgoals are well-aligned with human preferences. This temporal offset, where subgoals consistently lead actual execution by approximately one timestep forward, empirically validates the quality and effectiveness of our subgoal generation mechanism.

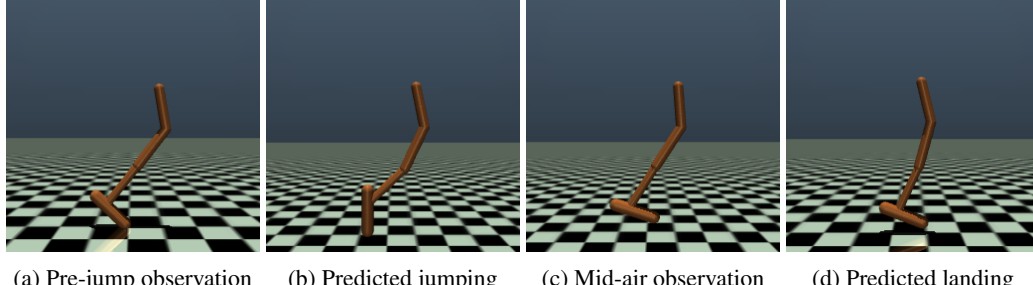

|(a) Pre-jump observation|(b) Predicted jumping|(c) Mid-air observation|(d) Predicted landing|

Figure 3: Qualitative analysis of subgoal extraction in the hopper environment. The predicted subgoals demonstrate forward-looking behavior: (a-b) optimal jumping configuration predicted during preparatory phase, and (c-d) landing-ready posture predicted during aerial phase. This temporal anticipation validates the predictive nature of our subgoal generation mechanism.

Table 4: Performance comparison between Preference Transformer and SPOT. The query number is different for each environment: hopper-medium-expert-v2 uses {100, 50, 30}, while walker2d-medium-replay-v2 uses {500, 100, 50}.

| Environment | Model | Number of Query | Score |
|---|---|---|---|
| hopper-medium-expert | Preference Transformer | 100 | $76.21 \pm 1.74$ |
| | | 50 | $75.55 \pm 2.12$ |
| | | 30 | $68.06 \pm 4.92$ |
| | SPOT | 100 | $99.37 \pm 8.35$ |
| | | 50 | $85.99 \pm 12.20$ |
| | | 30 | $85.09 \pm 8.54$ |
| walker2d-medium-replay | Preference Transformer | 500 | $73.64 \pm 2.12$ |
| | | 100 | $73.43 \pm 7.60$ |
| | | 50 | $71.98 \pm 4.93$ |
| | SPOT | 500 | $77.51 \pm 2.60$ |
| | | 100 | $75.87 \pm 2.03$ |
| | | 50 | $75.39 \pm 3.32$ |

## 5.5 QUERY EFFICIENCY

Another interesting benefit of SPOT is its query efficiency. We conducted comparative experiments across different query numbers and environments. The results in Table 4 demonstrate that SPOT achieves superior performance, generally in the hopper-medium-expert-v2 environment outperforming the preference Transformer. In the walker2d-medium-replay-v2 environment, both models showed consistent performance across varying query lengths, with our enhanced model maintaining stable scores around 75 even as queries decreased from 500 to 50. Even with a query length of 50, it maintains consistent performance, whereas the Preference Transformer shows a performance decline. This stability and outstanding performance validates our method that subgoal utilization through CVAE can enhance query efficiency by providing shaped rewards that effectively compensate for reduced preference queries.

## 6 CONCLUSION

**Summary** We present SPOT (Subgoal-based Policy Optimization through Attention Weight), a framework that mitigates extrapolation errors in offline preference-based reinforcement learning via preference-aligned subgoals. Our approach identifies critical decision points derived from attention weights as subgoals, uses these waypoints to shape rewards, thereby reducing extrapolation error. Not only does SPOT mitigate extrapolation error but it also outperforms conventional preference-based methods across diverse benchmarks, validating the efficacy of subgoals. Our findings establish a promising direction to advance realiability and practical applicability via integrating subgoals with offline PbRL.

**Limitation & Future work** While our approach is designed to complement an existing preference learning framework that provides state-level importance weights, we focus our validation on the offline setting. Given that offline learning scenarios present more challenging conditions due to their inherent instabilities and limited exploration capabilities, we specifically chose this setting to test our method's fundamental effectiveness. Although our approach could be extended to online preference learning frameworks such as Hindsight Prior Learning (Verma & Susa, 2024), we leave the exploration of these extensions as future work. Furthermore, our work assumes relatively clean preference labels following standard conventions in offline PbRL literature. Investigating robustness to noisy preferences,where annotators provide inconsistent or conflicting labels, represents an important direction for future work, particularly as real-world deployment scenarios may involve imperfect human feedback.

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

## A    DATASETS AND TASKS DETAIL

We conduct experiments on a variety of well-established offline datasets, each encompassing multiple tasks of differing complexities. Specifically, we utilize *Gym Mujoco Locomotion* benchmarks, *Robosuite (Robomimic)* manipulation tasks, and *Meta-World* manipulation tasks. Below, we detail each dataset along with the tasks we examine.

**Locomotion.** We employ several locomotion tasks from the D4RL benchmark, particularly focusing on *Hopper* and *Walker2d*. These tasks require controlling simulated robots in an optimal manner. In *Hopper*, the objective is to move a one-legged robot forward while balancing speed, energy usage, and stability. The *Walker2d* task requires a two-legged robot to walk forward, maximizing forward distance and survival while minimizing control costs. Both tasks present challenges in maintaining stability and forward progress. All datasets for these tasks are accessible via D4RL's provided APIs, licensed under *CC BY 4.0*.

**Robosuite Robotic Manipulation.** Robosuite offers diverse manipulation tasks featuring 7-DoF robotic arms. In our experiments, we focus on two specific environments, *lift* and *can*. The *lift* task involves grasping and lifting a cube, while the *can* task entails picking up a soft-drink can and placing it into a designated bin. These data are sourced from two distinct types of teleoperation: one proficient teleoperator (*ph*) and six teleoperators with varying skill levels (*mh*). The tasks are sparsely rewarded, providing non-zero feedback only upon successful task completion or relevant subgoals.

**Meta-World.** Meta-World is a popular suite of simulated robotic manipulation tasks, commonly executed with a Sawyer robotic arm. We focus on three tasks: *button press*, where the arm must press a button on a surface; *drawer open*, which requires the arm to pull a drawer open; and *plate slide*, where the goal is to push a plate into a slot or cabinet. Each task tests different aspects of manipulation, such as precision control, grasping, and coordinated motion.

**Preference Dataset.** We employ preference annotations to generate reward signals for the above tasks, following methods from offline preference-based Reinforcement Learning Kim et al. (2023); Hejna & Sadigh (2023). We utilized baseline implementations from publicly available repositories. Specifically, for D4RL and Robosuite, preference annotations were obtained using *Preference Transformer*[1], while for Meta-World tasks, preference data were derived from the IPL framework[2]. These annotations provide pairwise feedback on short segments of trajectories, enabling training of a reward model even in the absence of explicit numerical rewards.

## B    EXPERIMENTAL DETAILS

### B.1    ALGORITHM IMPLEMENTATIONS

we selects the six kinds of baselines which are mostly well-known and top-performing in offline preference based reinforcement learning fields. We are explaining the details about baselines and our methods.

#### B.1.1    PREFERENCE TRANSFORMER

The architecture of Preference Transformer (PT) consists of a causal transformer with a bidirectional self-attention mechanism. Following the original implementation, we employ a single-layer architecture with four self-attention heads, which provides an effective balance between computational efficiency and model performance. The transformer operates on an embedding dimension of 256, processing sequential data while maintaining temporal dependencies through its causal structure. This implementation entirely follows the original PT architecture Kim et al. (2023), ensuring reproducibility while maintaining computational tractability.

The complete hyperparameter configuration is detailed in Table 5 as depicted in Kim et al. (2023).

---

[1]https://github.com/csmile-1006/PreferenceTransformer
[2]https://github.com/jhejna/inverse-preference-learning

Table 5: Hyperparameters of Preference Transformer Kim et al. (2023)

| Hyperparameter | Value |
|---|---|
| Number of layers | 1 |
| Number of attention heads | 4 |
| Embedding dimension | 256 |
| (Casual transformer, Preference attention layer) | |
| Batch size | 256 |
| Dropout rate (embedding, attention, residual connection) | 0.1 |
| Learning rate | 0.0001 |
| Optimizer | AdamW Loshchilov & Hutter (2019) |
| Optimizer momentum | $\beta_1 = 0.9, \beta_2 = 0.99$ |
| Weight decay | 0.0001 |
| Warmup steps | 500 |
| Total gradient steps | 10K |

### B.1.2 BASELINE IMPLEMENTATIONS

For comprehensive evaluation, we implemented several baseline approaches. The Markovian Reward (MR) model follows the architecture specified in the original PT paper, utilizing their connected layer design for reward estimation. We maintained consistency with the PT implementation by adopting identical hyperparameter settings as detailed in the previous section.

We also compared our approach against Inverse Preference Learning (IPL), which is notable for its ability to operate without explicit reward modeling. For IPL implementation, we adhered to the original hyperparameter configuration as provided in their public repository. This ensures faithful reproduction of their reported methodology.

Additionally, we incorporated Human Preference Learning (HPL) as another baseline comparison. The implementation strictly follows the original authors' codebase[3], maintaining their specified hyperparameter settings to ensure accurate representation of their approach. This adherence to original implementations facilitates fair and reliable comparative analysis across different preference-based learning methods.

We further included Contrastive Preference Learning (CPL)[4] as a strong reward-learning baseline. For environments covered in the original CPL benchmark, we followed the default architecture and hyperparameter settings provided by the authors. Since CPL does not provide an implementation for robomimic (robosuite) tasks, we additionally implemented CPL on these domains by reusing the preference datasets released with PT and IPL. For these newly introduced robomimic (robosuite) tasks, we matched CPL's training hyperparameters to those used by our proposed method to ensure a fair comparison.

Finally, we evaluated In-Dataset Trajectory Return Regularization (DTR)[5] , which regularizes the policy toward the empirical return distribution of the offline dataset to mitigate reward bias and over-optimistic extrapolation from misspecified or noisy rewards. Because the original implementation does not cover robomimic (robosuite) or Meta-World, we extended DTR to these domains using the same preference datasets as PT and IPL, mirroring our CPL setup. We used the official hyperparameters for supported tasks, and for the newly added robomimic (robosuite) and Meta-World tasks we matched the hyperparameter configuration of our model for a controlled comparison.

### B.1.3 TRAINING DETAILS

Our codebase is implemented upon the same reimplemented GPT with JAX framework as in *Preference Transformer*. We utilize comparable hyperparameters throughout all experiments, including a segment length of 100 and a similar number of preference queries. we trained CVAE with hyperparameters listed in Table 6.

---

[3] http://github.com/typoverflow/WiseRL
[4] https://github.com/jhejna/cpl
[5] https://github.com/TU2021/DTR

Table 6: Hyperparameters for our CVAE

| Hyperparameter | Value |
|---|---|
| Dimension of latent variable $z$ | 16 |
| Hidden dimensions (encoder / prior network) | [32, 64, 32] |
| Learning rate | $1 \times 10^{-4}$ |
| Batch size | 256 |
| Posterior / prior distribution | Diagonal Gaussian |
| KL loss term weighting | 1.0 |
| training steps | 100k |
| output dim | observation dim |

For the IQL training, we apply a standard reward normalization process to ensure stable learning. And we use publicly release IQL setting followed by conventional researches. All experiments are conducted using JAX Bradbury et al. (2018) on a machine equipped with dual Intel Xeon E5-2630 v4 CPUs (20 physical cores, 40 threads) and a single NVIDIA GeForce RTX 1080 Ti GPU. We train both the learned reward model and IQL policy over 5 random seeds for each of the tasks. The total training time varies with the complexity of the environment; however, on average, each reward model requires only a few minutes, while the subsequent IQL training generally completes within an hour for each dataset. For the Ablation experiments, we utilized 3 random seed value to get performance results. This parallel training architecture enables computational efficiency by minimizing additional training overhead for each procedural step. For the ablation study, we conduct to visualize the correlation between ground truth, PT, and our methods. we sample 10K samples from dataset to visualize the distribution and compare the correlation and MSE value between each values.

## C  ABLATION STUDIES

### C.1  AUXILIARY LOSS FUNCTIONS

To evaluate the effectiveness of our auxiliary cosine similarity loss design in CVAE training, we conduct ablation studies comparing MSE loss alone against the combination of MSE with cosine similarity loss. Table 7 demonstrates that the combined auxiliary loss consistently outperforms MSE-only training across all tested environments. The cosine similarity component provides semantic informations that improve subgoal alignment, particularly benefiting manipulation tasks where spatial relationships are critical.

Table 7: Performance comparison of auxiliary loss functions

| Environment | MSE Only | MSE + Cosine Similarity |
|---|---|---|
| lift-mh | $48.07 \pm 16.25$ | $\mathbf{71.19 \pm 15.24}$ |
| can-mh | $39.22 \pm 3.53$ | $\mathbf{59.56 \pm 0.29}$ |
| walker2d-medium-expert-v2 | $109.23 \pm 0.25$ | $\mathbf{110.13 \pm 0.21}$ |
| walker2d-medium-replay-v2 | $73.61 \pm 1.23$ | $\mathbf{77.51 \pm 3.19}$ |

### C.2  COMPUTATION TIME

We conducted all experiments on a machine equipped with dual Intel Xeon E5-2630 v4 CPUs (20 physical cores, 40 threads) and a single NVIDIA GeForce RTX 1080 Ti GPU to ensure fair and reproducible comparisons. As shown in Table 8, we separately measured the computation time for reward model training and the offline RL phase. SPOT introduces only a small amout of computational overhead compared to PT—specifically, an additional 628 seconds (10.5 minutes) for hopper-m-e and 1,674 seconds (27.9 minutes) for walker-m-r. This marginal increase is negligible when weighed against the substantial performance improvements SPOT delivers. In stark contrast, HPL requires nearly 7× more computation than SPOT (37,085s vs. 5,824s on hopper-m-e), making

it prohibitively expensive for practical deployment. Even IPL, which eliminates reward model training entirely, takes 2.7× longer than SPOT due to its inefficient policy learning phase. While SPOT does introduce an additional reward model training stage compared to IPL, the resulting performance gains—combined with faster convergence during offline RL—more than justify this investment. The total wall-clock time remains competitive with simpler baselines while achieving superior sample efficiency and final performance.

Table 8: Training Time Comparison Between Methods

| Method | Environment | Reward Model Training | Offline RL | Total Time |
|--------|-------------|----------------------|------------|------------|
| IPL | hopper-m-e | – | 4:19:07 (15,547s) | 4:19:07 (15,547s) |
| HPL | hopper-m-e | 4:30:37 (16,237s) | 5:47:28 (20,848s) | 10:18:05 (37,085s) |
| CPL | hopper-m-e | 0:19:03 (1,143s) | 1:14:21 (4,462s) | 1:33:24 (5,605s) |
| PT | hopper-m-e | 0:22:54 (1,374s) | 1:03:42 (3,822s) | 1:26:36 (5,196s) |
| SPOT | hopper-m-e | 0:31:18 (1,878s) | 1:05:45 (3,946s) | 1:37:03 (5,824s) |
| IPL | walker-m-r | – | 4:03:06(14,586s) | 4:03:06(14,586s) |
| HPL | walker-m-r | 4:40:35 (16,835s) | 5:49:07 (20,947s) | 10:29:42 (37,782s) |
| CPL | walker-m-r | 1:20:47 (4,847s) | 1:18:36 (4,715s) | 2:39:23 (9,562s) |
| PT | walker-m-r | 1:44:17 (6,257s) | 1:02:59 (3,780s) | 2:47:16 (10,037s) |
| SPOT | walker-m-r | 2:07:14 (7,634s) | 1:07:57 (4,077s) | 3:15:11 (11,711s) |

## C.3 ADDITIONAL VISUALIZATION AND QUALITY ASSESSMENT

To evaluate the quality and distribution of generated subgoals, we conduct comprehensive visualization analysis using multiple dimensionality reduction techniques and quantitative metrics. Figure 4 presents t-SNE, PCA, and UMAP projections of observation embeddings (blue) and subgoal embeddings (red) in the hopper-medium-expert environment. The t-SNE visualization reveals a complex, nonlinear manifold structure where subgoal embeddings exhibit strategic clustering in specific regions, indicating that our CVAE effectively identifies critical state transitions and key decision points in the task space. The PCA projection demonstrates a more linear trajectory pattern with observations forming a distinct path from upper left to lower right regions, while subgoal embeddings concentrate at strategically important locations, particularly at trajectory endpoints. The UMAP visualization provides complementary insights into the local neighborhood structure, confirming that generated subgoals maintain semantic consistency with the observation space while focusing on pivotal states.

To quantitatively assess the quality of our subgoal generation, we employ precision and recall metrics following the methodology of (Kynkäänniemi et al., 2019). Our evaluation on the hopper-medium-expert-v2 environment yields a precision of 0.652 and recall of 0.795, indicating that our CVAE generates high-quality subgoals with good coverage of the target distribution while maintaining reasonable fidelity. The high recall value demonstrates that our method successfully captures the diversity of critical states in the expert demonstrations, while the balanced precision score confirms that generated subgoals avoid spurious or irrelevant states, validating the effectiveness of our subgoal extraction mechanism.

## C.4 SUBGOAL EXTRACTION IN GOAL-ORIENTED MANIPULATION TASKS

To further validate the generalizability of our subgoal extraction mechanism, we present a qualitative analysis on the Can-Ph goal-oriented manipulation task in Figure 5. We examine the correspondence between original observations and predicted subgoals across three critical phases of the manipulation sequence. During the pre-contact phase (Figures 5a–5d), while the agent is still approaching the target object, the predicted subgoal already displays successful grasping with proper gripper engagement. This forward-looking behavior provides the agent with a clear intermediate objective prior to physical contact. In the subsequent reaching phase (Figures 5b–5e), as the robot arm transports the object toward the target location, the corresponding subgoal projects a more advanced state with the object positioned closer to the goal region, effectively guiding trajectory planning. Finally, during the drop preparation phase (Figures 5c–5f), when the agent is maneuvering to release the ob-

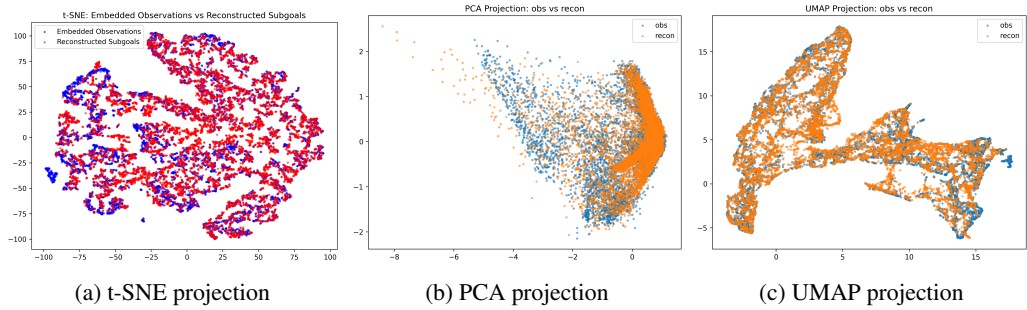

(a) t-SNE projection     (b) PCA projection     (c) UMAP projection

Figure 4: Latent space visualization of observation embeddings (blue) and subgoal embeddings (red) in hopper-medium-expert environment using different dimensionality reduction techniques. All visualizations demonstrate strategic clustering of subgoals at critical decision points while maintaining semantic consistency with the observation space.

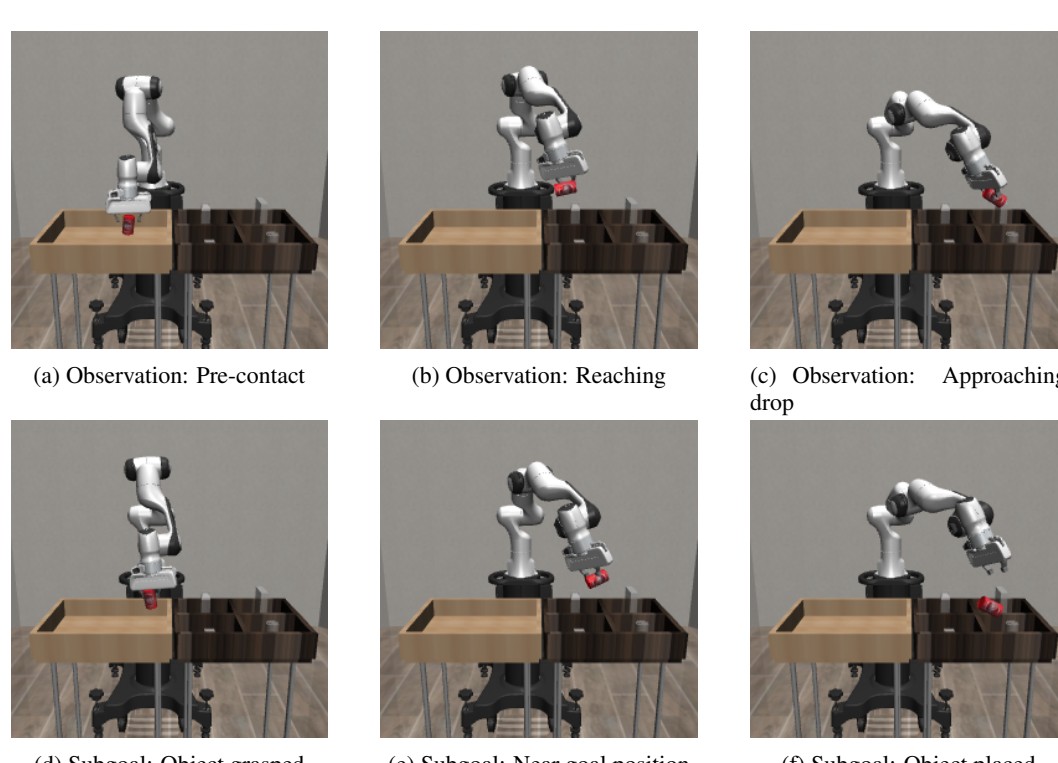

(a) Observation: Pre-contact     (b) Observation: Reaching     (c) Observation: Approaching drop

(d) Subgoal: Object grasped     (e) Subgoal: Near goal position     (f) Subgoal: Object placed

Figure 5: **Subgoal extraction on the Can-Ph manipulation task.** Original observations (top row) and their corresponding predicted subgoals (bottom row) across three critical phases: (a–d) pre-contact phase where the agent approaches the target, with the subgoal anticipating successful grasping; (b–e) reaching phase where the robot transports the object toward the target location, with the subgoal projecting closer proximity to the goal region; (c–f) drop preparation phase, with the subgoal displaying the completed placement.

ject, the subgoal already exhibits the completed placement in the target location. These qualitative results verify our findings from Section 5.4, demonstrating that the subgoal extraction mechanism consistently maintains a one-timestep forward-looking temporal offset across diverse task domains, including goal-oriented manipulation tasks. Notably, the predicted subgoals preserve fine-grained spatial relationships—including robot configuration, object poses, and scene geometry—while simultaneously projecting meaningful progress toward task completion. This visualization validates

---

**Algorithm 1** SPOT: Attention-Driven Subgoal Learning and Reward Shaping

---

1: **Input:** offline dataset $\mathcal{D}$, preference dataset $\mathcal{D}_{pref}$, percentile $K$, KL weight $\beta$, shaping weight $\lambda$

2: **Output:** reward model $r_{\text{model}}$, subgoal CVAE $(\phi, \theta, \psi)$, offline policy $\pi$

3: **// Stage 1: Subgoal Learning Via CVAE**

4: Initialize reward model $r_{\text{model}}$, encoder $\phi$, decoder $\theta$, prior $\psi$

5: Initialize subgoal triplet buffer $S_g(\sigma; K) \leftarrow \emptyset$

6: **while** not converged **do**

7:     Sample preference minibatch of trajectories $\{\sigma_{pref}, \sigma_{unpref}\} \subset \mathcal{D}_{pref}$

8:     Update Preference Transformer and $r_{\text{model}}$ on preference pairs from $\{\sigma_{pref}, \sigma_{unpref}\}$

9:     **for** each preferred trajectory $\sigma_{pref} = \{(s_t, a_t)\}_{t=1}^{T}$ in the minibatch **do**

10:         Compute attention weights $w_t$ and reward estimates $\hat{r}_t = r_{\text{model}}(s_t, a_t)$

11:         Compute $\bar{r}(\sigma) = \frac{1}{T} \sum_t \hat{r}_t$ and threshold $\alpha_K(\sigma)$ as the $(1 - K)\%$ quantile of $\{w_t\}$

12:         Define subgoal set $S_g(\sigma; K) = \{s_t \mid w_t \geq \alpha_K(\sigma),\ \hat{r}_t \geq \bar{r}(\sigma)\}$

13:         Construct triplets $(s_t, a_t, g_t)$ with $g_t \in S_g(\sigma; K)$ and append to $\mathcal{D}_{\text{sub}}$

14:     **end for**

15:     **if** $\mathcal{D}_{\text{sub}}$ is not empty **then**

16:         Sample minibatch $(s_t, a_t, g_t) \sim S_g$

17:         Encode $q_\phi(z \mid g_t, s_t, a_t)$, prior $p_\psi(z \mid s_t, a_t)$, sample $z$, decode $\hat{g}_t \sim p_\theta(g \mid z, s_t, a_t)$

18:         Compute $L_{\text{CVAE}}$ by Eq. (7)

19:         Compute cosine similarity $c_t = \frac{\hat{g}_t^\top g_t}{\|\hat{g}_t\|_2 \|g_t\|_2}$

20:         Compute $L_{\text{sim}} = -\frac{1}{2}(1 + c_t)$ and $L_{\text{total}} = L_{\text{CVAE}} + L_{\text{sim}}$

21:         Update $\phi, \theta, \psi$ by a gradient step on $L_{\text{total}}$

22:     **end if**

23: **end while**

24: **// Stage 2: Offline RL with subgoal-guided shaping**

25: Initialize offline RL policy $\pi$

26: **while** not converged **do**

27:     Sample batch $(s_i, a_i, s_i')$ from $\mathcal{D}$

28:     Generate subgoals $\hat{g}_i$ from CVAE given $(s_i, a_i)$

29:     Compute $\text{sim}_i = \frac{s_i'^\top \hat{g}_i}{\|s_i'\|_2 \|\hat{g}_i\|_2}$ and $r_{\text{shape}}(i) = \frac{1}{2}(1 + \text{sim}_i)$

30:     Compute $r_{\text{final}}(i) = r_{\text{model}}(s_i, a_i) + \lambda\, r_{\text{shape}}(i)$

31:     Update policy $\pi$ using offline RL (e.g., IQL) with rewards $r_{\text{final}}$

32: **end while**

---

that our learned latent representation captures task-relevant structure essential for effective subgoal-based reward shaping in complex manipulation scenarios.

## C.5 PSEUDO CODE

For completeness, we provide pseudo-code of the overall algorithm in Algorithm 1.

