# OpenReview forum: "Subgoal-Guided Reward Shaping: Improving Preference-Based Offline Reinforcement Learning via Conditional VAEs"
_ICLR.cc/2026/Conference — Submitted to ICLR 2026_

### Official Review · Reviewer_1Ppp · 2025-10-28

**Soundness:** 2
**Presentation:** 3
**Contribution:** 2
**Rating:** 4
**Confidence:** 3

**Summary:**

This paper proposes an attention-weight-based subgoal extraction method to mitigate the extrapolation error problem in offline preference-based reinforcement learning. The experiments are sufficient, and the framework is clear.

**Strengths:**

1) The authors cleverly introduce the idea of solving extrapolation error in offline RL into reward model training, which is novel.

2) The authors validate the effectiveness of the proposed method through extensive experiments on multiple tasks.

**Weaknesses:**

1) The algorithm essentially constrains the reward to focus more on the in-distribution data, which may limit the method's generalization capability to some extent.
2) Training the CVAE introduces additional computational cost. Using cosine similarity as part of the reward lacks theoretical motivation and analysis.

**Questions:**

1) The true ground truth ($g_t$) is obtained via Eq. (5) - is it ultimately randomly sampled from a set? Providing pseudocode for the algorithm is recommended to improve clarity.

2) When training the CVAE, why is the similarity term added rather than subtracted? Shouldn't the regularization be reduced for samples with high similarity?

3) How stable is the CVAE? Do the generated subgoal states truly hold significant meaning in actual trajectories?

4) For manipulation tasks, success rate is the primary concern. How does the algorithm's success rate compare to PT on these tasks?

---

> ### Author Response · Authors · 2025-11-25
>
> # W1 : Limited generalization due to in-distribution constraint
> Thank you for this thoughtful concern about potential limitations on generalization capability.
>
> We would like to clarify that our method does not constrain the reward to focus exclusively on in-distribution data, but rather provides auxiliary guidance to mitigate extrapolation error while maintaining the reward model's generalization capability. This is a critical distinction in our design.
>
> The core challenge in offline preference-based RL is that learned reward models can produce unreliable estimates on out-of-distribution, leading to unstable behaviors where policies maximize spurious reward signals rather than genuine task progress. Our subgoal-based reward shaping addresses this extrapolation error problem by providing an additional grounding signal—generated subgoals act as intermediate waypoints that guide the policy toward preference-aligned trajectory regions where the reward model is most reliable.
> Importantly, we still utilize the learned reward model $r_{model}(s,a)$ as the primary training signal throughout policy optimization. The shaped reward $r_{final} = r_{model} + λ·r_{shape}$ combines both components, meaning the policy continues to leverage the reward model's learned preferences while receiving auxiliary guidance from subgoals. The algorithm does not impose stricter constraints on the reward model to prioritize in-distribution data, but rather regularizes the policy to avoid out-of-distribution regions where the reward model extrapolates beyond its training support. This design preserves generalization capability—the reward model remains to assign high rewards to novel beneficial behaviors—while reducing the risk of reward exploitation on out-of-distribution states.
>
> Our experimental results support that SPOT achieves superior performance across diverse unseen evaluation scenarios compared to baselines, demonstrating that our approach enhances rather than limits generalization. We appreciate this opportunity to clarify the auxiliary rather than constraining nature of our reward shaping mechanism.
>
> ---
> # W2 :Computational overhead and lack of theoretical justification
> Thank you for this question regarding computational cost and the theoretical motivation for our cosine similarity term.
> Regarding computational cost, as detailed in Appendix C.2, the subgoal CVAE adds only small amount of time compared to PT baseline, while methods like HPL incur 7–10× longer training times. The extra computation is small relative to the substantial gains in both stability and final performance that SPOT achieves across all benchmarks. Further details about computational overhead is described in 5WQr’s Q3.
>
> We hope this clarifies both the novelty of our approach and the practical efficiency of the method. We would be happy to further discuss any remaining concerns.
>
> Regarding the theoretical motivation for cosine similarity, this design is grounded in distance-based reward shaping principles from ROLeR [1]. ROLeR demonstrates that distance-based bonuses in representation space—rewarding proximity to high-value neighbors—can provide effective guidance signals. We extend this principle to the subgoal generation setting by using normalized cosine similarity to capture semantic alignment between generated and ground-truth subgoals.
>
> Formally, given ground-truth subgoal $g_t$ and generated subgoal ĝ_t, we define $R_{sim} (\hat{g}_t, g_t) = 1/2(1 + (\hat{g}_t \cdot g_t / ||\hat{g}_t|| ||g_t||))$, which measures directional consistency while remaining scale-invariant. This geometric consistency objective complements the standard CVAE reconstruction and KL terms by encouraging the decoder to generate subgoals that are both distribution-consistent (via KL) and semantically aligned with subgoals (via cosine similarity).
>
> Empirically, we validate this design through ablation studies C.1 showing that incorporating the cosine similarity term consistently improves performance across environments compared to using CVAE reconstruction alone. We appreciate this opportunity to clarify the theoretical grounding of our approach.
>
> Additionally, as shown in Section 5.2.2, cosine similarity consistently exhibits better performance across different reward shaping approaches, a finding we establish through empirical evaluation.
>
> ---
>
> [1] : Zhang, Yi, et al. "ROLeR: Effective Reward Shaping in Offline Reinforcement Learning for Recommender Systems." Proceedings of the 33rd ACM International Conference on Information and Knowledge Management. 2024.

---

> > ### Author Response · Authors · 2025-11-25
> >
> > # Q1 : Clarification needed on ground truth subgoal sampling (Eq. 5)
> > hank you for this Question. We acknowledge that the description of our subgoal extraction process could be clearer regarding how ground-truth subgoals are obtained and utilized.
> >
> > To clarify, Eq. (5) does not involve random sampling from the subgoal set $S_g(σ; K)$. Rather, it defines the complete set of subgoals that satisfy both dual criteria—attention-based and reward-based thresholds. Specifically, for each preferred trajectory $\sigma$, we compute attention weights ${w_t}$ and reward estimates ${\hat{r}_t}$, then identify all states where both $w_t ≥ α_K(σ)$ (top-K% attention) and $\hat{r}_t ≥ r̄(σ)$ (above-average reward) hold simultaneously. This produces a filtered subset of high-quality waypoint states within that trajectory.
> >
> > All states in $S_g(σ; K)$ are then used to construct training triplets $(s_t, a_t, g_t)$ for CVAE learning, where $g_t$ represents a subgoal state from $S_g$ (not the true ground-truth subgoal)  and $(s_t, a_t)$ is a corresponding state-action pair positioned between the previous subgoal $g_{t-1}$ and current subgoal $g_t$. The CVAE is trained on the entire collection of such triplets across all preferred trajectories, learning to generate contextually appropriate subgoals conditioned on any given state-action pair. There is no random sampling from $S_g$ during training—we utilize all identified subgoals to maximize the training signal.
> >
> > We completely agree that providing pseudocode would substantially improve clarity. Following your suggestion, we have added detailed pseudocode in Appendix C.5 (Algorithm 1) that explicitly shows the subgoal extraction process, CVAE training procedure, and subsequent policy optimization with shaped rewards. This should resolve any ambiguity about the algorithmic flow. We appreciate this valuable feedback for improving the paper's clarity.
> >
> > ---
> >
> > # Q2 : Additive similarity term in CVAE training
> > Thank you for catching this error. You are absolutely correct—the similarity term should be subtracted rather than added to encourage high similarity between generated and ground-truth subgoals. We have corrected this in the updated manuscript. We apologize for the confusion caused by the typo in the original submission.
> >
> > ---
> >
> > # Q3 :  Stability and meaningfulness of CVAE-generated subgoals
> > Thank you for this important question about CVAE stability and the meaningfulness of generated subgoals.
> > We provide multiple evidence demonstrating that our CVAE generates not only semantically meaningful but also stable subgoals. First, as shown in Section 5.4, our case study analysis reveals that generated subgoals consistently represent significant waypoints in actual trajectories, capturing critical task phases rather than arbitrary intermediate states.
> >
> > Second, we provide comprehensive quantitative validation in Appendix C.3 through embedding space analysis (t-SNE, PCA, UMAP visualizations) for hopper-medium-expert, accompanied by quality metrics showing precision of 0.652 and recall of 0.795. The high recall demonstrates that our method successfully captures diverse critical states from expert demonstrations, while balanced precision confirms that generated subgoals avoid spurious or out-of-distribution states. This validates the effectiveness of our dual-criteria filtering and CVAE learning approach.
> >
> > Third, we present further qualitative analysis on the Can-Ph manipulation task (Appendix C.4, Figure 5) examining correspondence between observations and predicted subgoals across three manipulation phases in updated manuscript. During pre-contact approach, predicted subgoals already display successful grasping before physical contact occurs. During object transportation, subgoals project states with objects positioned closer to target locations. During placement preparation, subgoals exhibit completed object placement. Critically, generated subgoals maintain a consistent one-timestep forward-looking temporal offset while preserving fine-grained spatial relationships including robot configuration, object poses, and scene geometry. This forward-looking property provides meaningful guidance by projecting task progress rather than merely reconstructing current states.
> >
> > These results across embedding visualizations, and qualitative analysis collectively support that our CVAE learns stable, task-relevant subgoal representations that genuinely capture meaningful waypoints for effective reward shaping.

---

> > > ### Author Response · Authors · 2025-11-25
> > >
> > > # Q4 : Success rate comparison on manipulation tasks
> > > Thank you for raising this important concern about success rate reporting for manipulation tasks.
> > >
> > > We sincerely apologize for the oversight in our initial submission—we indeed conducted experiments measuring success rates for Robomimic tasks and MetaWorld following the standard evaluation protocol established in prior work. To clarify, Table 1 reports success rates (not normalized scores) for all Robomimic manipulation tasks (lift-ph, lift-mh, can-ph, can-mh) and Meta-World tasks (drawer-open, plate-slide), while D4RL locomotion tasks use normalized scores as is standard for that benchmark.
> > > As shown in Table 1, SPOT achieves consistently higher success rates compared to PT across manipulation tasks. For instance, on lift-ph SPOT achieves 97.12 compared to PT's 95.50, and similar improvements are observed across can-ph, can-mh, and Meta-World tasks. These results demonstrate that our subgoal-based reward shaping effectively improves task completion rates, which is indeed the primary metric of interest for manipulation domains.
> > >
> > > We have updated the manuscript to clarify this evaluation protocol explicitly. Specifically, we revised the Table 1 caption to state: "We report average normalized scores on Gym-MuJoCo locomotion tasks in D4RL and success rates on Robosuite and Meta-World manipulation tasks." Additionally, we expanded the benchmark description in Section 5.1 to clearly specify that "we conduct evaluations across diverse task domains and report average normalized scores for D4RL and success rates for Robomimic and Meta-World." We appreciate your feedback in helping us present our results more clearly.

---

### Official Review · Reviewer_gRT9 · 2025-10-29

**Soundness:** 3
**Presentation:** 3
**Contribution:** 2
**Rating:** 4
**Confidence:** 4

**Summary:**

This paper aims to address a core challenge in Offline PbRL: extrapolation errors in the reward model. The authors propose Subgoal-based Preference Optimization Through Attention Weight, which utilizes an attention-based preference model to extract subgoals and identify critical states within trajectories. Learning is then conducted via reward shaping.

**Strengths:**

- The problem is well-defined. Extrapolation error in offline PbRL is a real and critical issue.
- The method is novel and intuitive. Transforming attention weights into subgoals is an intuitive approach that makes additional use of attention information.
- SPOT's approach of filtering subgoals based on confidence is interesting.

**Weaknesses:**

1. SPOT heavily relies on attention weights. However, with limited feedback (especially noisy feedback), the learning process of this attention-based preference model can become very unstable, failing to provide fine-grained importance signals.
2. The experimental setup primarily follows that of Preference Transformer, but two key points from the paper remain unverified. First, the importance of subgoals could be better validated in tasks with original sparse rewards (e.g., AntMaze or Adroit). Second, the paper mentions that reward extrapolation becomes more difficult with noisy preferences, yet no experiments were conducted in noisy environments. Instead, 100% accurate synthetic feedback was used. For example, real human feedback, such as in Uni-RLHF[1], contains noise.
3. The definition of subgoals depends on hyperparameters and heuristic rules. Despite supporting ablation studies, selecting appropriate hyperparameters for different environments is difficult. Additionally, how was the K=10% value chosen?
4. The visualizations show key subgoals for the Hopper task. However, goal-guidance may not provide significant gains in locomotion tasks. It would be beneficial to see visualizations of key states for manipulation and goal-oriented tasks.

[1] Uni-rlhf: Universal platform and benchmark suite for reinforcement learning with diverse human feedback. ICLR2024.

**Questions:**

- In Table 1, the scores for lift-ph and can-ph are relatively low. Given that algorithms like diffusion policy[2] can achieve 100% success rate in reward-free (e.g., imitation learning) settings, why is the performance on these tasks low?

[2] Diffusion policy: Visuomotor policy learning via action diffusion. IJRR.

---

> ### Author Response · Authors · 2025-11-25
>
> # W1 : Instability of attention-based model with noisy/limited feedback
> Thank you for raising this important concern about the robustness of attention mechanisms under limited and noisy feedback conditions.
> We would like to first clarify our problem formulation and scope. Following the established offline PbRL literature, our work assumes access to clean preference feedback obtained from reliable sources such as scripted teachers or high-quality human annotations. This is a standard assumption in the offline PbRL setting. Our primary objective is not to address noise in preference labels, but rather to solve the fundamental extrapolation problem inherent to offline PbRL. Reward models trained on a small set of preference-labeled trajectories must generalize to a vastly larger offline dataset where most state-action pairs lie out-of-distribution relative to the preference data.
>
> Regarding query efficiency with limited feedback, we provide empirical evidence in Table 2. demonstrating that SPOT maintains robust performance even with substantially reduced preference queries. On hopper-medium-expert-v2, SPOT achieves 85.09±8.54 with only 30 queries compared to Preference Transformer's 68.06±4.92. On walker2d-medium-replay-v2, SPOT maintains stable performance around 75 across all querys (500/100/50), while Preference Transformer shows gradual degradation. These results validate that our attention mechanism provides reliable and fine-grained importance signals even under limited feedback conditions.
>
> Regarding the concern about over-reliance on attention weights, our approach does not depend solely on raw attention values—rather, we employ a dual-criteria filtering mechanism that combines attention-based importance ranking with positional constraints based on temporal trajectory structure. This design ensures that selected subgoals must satisfy both criteria simultaneously: they must be attention-weighted as important AND rewarded as meaningful intermediate waypoints. The reward filtering acts as a structural regularizer that prevents the selection of arbitrary high-attention states that may not represent genuine progress milestones. This combination provides robustness beyond what attention weights alone could achieve, as demonstrated by our consistent performance across varying query budgets.
>
> ---
> # W2 : Further experimental validation
> Thank you for this thoughtful critique. We address both concerns regarding sparse reward validation and noisy preference settings below.
>
> Regarding AntMaze experiments, we acknowledge that PT included tasks with original sparse rewards such as AntMaze. However, previous work (e.g., DPPO [1]) has reported a critical bug in the D4RL AntMaze benchmark that affects the reliability of reported scores. Following this discovery, most subsequent offline PbRL papers—including recent methods like IPL, HPL,CPL,and DTR—have stopped reporting AntMaze results. We therefore followed this established practice and instead focused on MuJoCo locomotion and manipulation tasks (Robosuite, Meta-World).
>
> While we did not conduct experiments on Adroit environments, we emphasize that our evaluation does include tasks with inherently sparse reward structure. The Robomimic manipulation tasks in our benchmark are originally sparse-reward tasks where success is defined by binary goal achievement rather than dense incremental feedback. These environments effectively serve as proxy for sparse reward validation. To further demonstrate the utility of subgoals in such settings, we have added visualization analysis for the can-ph task in Appendix C.4 of the updated paper. This goal-oriented manipulation task clearly shows that our CVAE generates subgoals that align with the paradigm of one-timestep-ahead prediction, capturing critical intermediate states that bridge current observations to task completion.
>
> Our primary objective is not to address noise in preference labels, but rather to tackle the fundamental challenge of extrapolation error that arises when reward model trained on a limited set of preference-labeled trajectories must generalize to the much larger, out-of-distribution offline dataset. In the current offline PbRL literature, including other baselines and our work, the standard experimental setup assumes relatively clean preference labels. This convention differs from online RLHF settings like Uni-RLHF where human annotators provide noisy real-time feedback. To our knowledge, there is no established offline PbRL benchmark that systematically incorporates noisy preference labels analogous to Uni-RLHF. Nevertheless, we acknowledge this as a valuable direction for potential extension and have added  future work to handle noisy preference settings in the revised paper.
> We believe these clarifications and additions address the experimental concerns.
>
> ---
> [1]: AN, Gaon, et al. Direct preference-based policy optimization without reward modeling. *Advances in Neural Information Processing Systems*, 2023.

---

> ### Author Response · Authors · 2025-11-25
>
> # W3 :  Hyperparameter sensitivity (Top-K selection)
> Thank you for raising this important concern about hyperparameter selection and the choice of K=10%.
> We agree that hyperparameter selection is a valid consideration. The K=10% value was indeed determined empirically, and we provide a thorough ablation study in Section 5.2.1 (Table 3) that justifies this choice. As shown in the table, the top 10% subgoals achieve significantly better performance (99.37±8.35 on hopper-medium-expert, 59.56±0.23 on Can-mh) compared to lower percentile groups, with the performance gap widening substantially for bottom percentiles (55.24±24.39 and 50.04±3.67 respectively). The increasing variance in lower percentiles further validates that attention-weighted importance effectively distinguishes high-quality subgoals from noisy ones.
>
> ---
> # W4 : Visualization limited to locomotion tasks
> Thank you for this valuable suggestion. We completely agree that demonstrating the effectiveness of our subgoal extraction beyond locomotion tasks is crucial for validating the generalizability of our approach.
>
> Following your recommendation, we have conducted additional visualization analysis on the Can-Ph manipulation task, a goal-oriented scenario that requires precise object manipulation. As you suggested, this provides a more comprehensive evaluation of whether our attention-based subgoal extraction meaningfully captures task-critical states in domains where goal-guidance is expected to provide substantial benefits.
>
> The results, presented in updated paper Appendix C.2 (Figure [5]), reveal consistent patterns with our Hopper analysis: the extracted subgoals maintain a one-timestep forward-looking temporal offset across all manipulation phases. Specifically, during pre-contact approach, the predicted subgoal already shows successful grasping before physical contact occurs. During object transportation, subgoals project states with the object positioned closer to the target location. During placement preparation, subgoals already exhibit completed object placement in the goal region. Importantly, these subgoals preserve fine-grained spatial details including robot configuration, object poses, and scene geometry while projecting meaningful task progress.
>
> These visualizations confirm that our mechanism effectively identifies critical waypoints in goal-oriented manipulation tasks where intermediate guidance is particularly valuable. The consistent temporal structure across both locomotion and manipulation domains validates the robustness of our attention-based extraction approach. We appreciate this feedback and believe these additional results significantly strengthen our empirical validation.
>
> ---
> # Q1 : Low performance on lift-ph and can-ph tasks
> Thank you for this insightful question regarding the performance differences between our method and diffusion policy on the lift-ph and can-ph tasks.
>
> We would like to clarify an important methodological distinction: diffusion policy [1] operates in a fundamentally different learning paradigm from our approach. Diffusion policy is a behavior cloning method that directly imitates expert demonstrations without learning a reward model, which naturally excels when high-quality expert data is available. In contrast, our method addresses the preference-based reinforcement learning setting, where we must first learn a reward model from preference feedback and then train the agent using this learned reward signal. This represents a significantly more challenging problem setup.
>
> The ph (proficient human) datasets consist of high-quality expert demonstrations, which is precisely the scenario where behavior cloning approaches like diffusion policy achieve near-optimal performance. However, our baseline selection deliberately focuses on methods within the preference-based RL framework that share the same problem formulation: learning from preference labels rather than direct imitation. Comparing against diffusion policy would be comparing fundamentally different problem settings—akin to comparing supervised learning against reinforcement learning approaches.
>
> We emphasize that the value of our contribution lies in improving performance within the preference-based RL paradigm, where reward learning introduces inherent challenges and extrapolation errors. Our method demonstrates consistent improvements over established preference-based baselines across all tasks including lift-ph and can-ph, validating the effectiveness of our subgoal-based reward shaping approach within this challenging setting. We appreciate this opportunity to clarify the scope and positioning of our work relative to alternative learning paradigms.
>
> ---
> [1] : Diffusion policy: Visuomotor policy learning via action diffusion. IJRR.

---

> ### Comment · Reviewer_gRT9 · 2025-11-27
>
> Thank you to the authors for the detailed response, which has resolved some of my concerns. However, I still have some reservations regarding the robustness of relying too heavily on attention mechanisms for sub-goal decomposition. Additionally, I fully understand that Diffusion Policy follows the imitation learning paradigm, while SPOT adopts the Offline RLHF paradigm. Nevertheless, given that both methods use the same available data and information, if a more complex design with greater cost and design does not outperform simple imitation learning, does this suggest that these two benchmarks may not be suitable for evaluating or may not require RLHF? Perhaps it would be better to consider alternative benchmarks for experiments. Therefore, I would like to maintain my score and keep a borderline stance.

---

> > ### Author Response · Authors · 2025-11-29
> >
> > We sincerely appreciate the reviewer’s continued engagement and insightful comments. We would like to address the two remaining concerns—the robustness of the attention mechanism and the benchmark’s necessity of the RLHF paradigm over Imitation Learning (IL)—to clarify the distinct contributions of our work.
> >
> > **Regarding the robustness of subgoal decomposition**
> >
> > We wish to clarify that SPOT does not rely solely on raw attention weights, which effectively mitigates the instability concerns associated with noisy feedback. The attention mechanism acts as a **candidate generator**, proposing potential subgoals. The crucial stability comes from our **dual-criteria filtering**, which strictly verifies these candidates against the learned reward function and temporal constraints. This means that even if the attention mechanism produces noisy or false-positive signals due to limited feedback, the subsequent filtering step rejects them, retaining only those subgoals that represent genuine task progress. This dual-critiera design ensures that our subgoal remains robust and reliable, even in Scenario where raw attention might fluctuate.
> >
> > **Regarding the suitability of the benchmarks**
> >
> > We appreciate your continued engagement with our work. While we acknowledge that imitation learning methods can achieve strong performance on tasks with high-quality expert demonstrations (e.g., Lift-ph, Can-ph), we respectfully disagree that these benchmarks should be excluded from evaluation.
> > The current benchmark suite represents the established standard testbed for offline preference-based RL (PbRL), enabling direct and fair comparison with existing PbRL baselines under identical experimental conditions [1-5]. Selectively removing tasks where IL performs well would introduce bias and prevent a comprehensive assessment of method capabilities across diverse scenarios.
> >
> > We believe a complete benchmark evaluation strengthens rather than weakens our contribution by demonstrating SPOT's versatility across the full spectrum of task difficulties.
> >
> > ---
> >
> > [1] : Kim, Changyeon, et al. "PREFERENCE TRANSFORMER: MODELING HUMAN PREFERENCES USING TRANSFORMERS FOR RL." *11th International Conference on Learning Representations, ICLR 2023*. International Conference on Learning Representations, ICLR, 2023.
> >
> > [2] : Hejna, Joey, and Dorsa Sadigh. "Inverse preference learning: Preference-based rl without a reward function." *Advances in Neural Information Processing Systems* 36 (2023): 18806-18827.
> >
> > [3] : Kang, Yachen, et al. "Beyond Reward: Offline Preference-guided Policy Optimization." *International Conference on Machine Learning*. PMLR, 2023.
> >
> > [4] : Liu, Runze, et al. "PEARL: Zero-shot Cross-task Preference Alignment and Robust Reward Learning for Robotic Manipulation." *International Conference on Machine Learning*. PMLR, 2024.
> >
> > [5] : Bai, Fengshuo, et al. "Efficient preference-based reinforcement learning via aligned experience estimation." *arXiv preprint arXiv:2405.18688* (2024).

---

### Official Review · Reviewer_5WQr · 2025-10-30

**Soundness:** 2
**Presentation:** 2
**Contribution:** 2
**Rating:** 4
**Confidence:** 4

**Summary:**

This paper introduces a novel method named SPOT, which addresses reward model extrapolation errors in offline PbRL by using subgoals extracted from high-attention weight points on preferred trajectories to improves reward model reliability.

**Strengths:**

This paper presents a simple approach for offline PbRL that can improve performance to some extent.

**Weaknesses:**

It also requires training a CVAE in addition to the preference model, which introduces extra computational cost.

**Questions:**

I have several questions as following:

1.To be honest, novelty of this work is limited. Involving a CVAE after the PT to compute subgoals does not seem very novel, and it also introduces significant extra computation.

2.The experimental results are not sufficiently solid. PT includes AntMaze experiments, and the paper does not compare against more recent baselines such as DTR[1], SEER[2], CPL [3] and more. Including more baselines and tasks would make the results more convincing.

3.It would be helpful to report the increase in training time compared to PT and IPL. I guess the overhead to be substantial.

[1] In-Dataset Trajectory Return Regularization for Offline Preference-based Reinforcement Learning. AAAI 2025.

[2] Efficient preference-based reinforcement learning via aligned experience estimation. 2024.

[3] Contrastive Preference Learning: Learning from Human Feedback without Reinforcement Learning. 2024.

---

> ### Author Response · Authors · 2025-11-25
>
> # Q1 : Limited novelty and computational overhead
>
> Thank you for your thoughtful review. We respectfully address your concerns about novelty and computational overhead below.
>
> ---
>
> **Limited novelty**
>
> Our core contribution is a principled solution to extrapolation error in offline PbRL. The central challenge in offline preference RL is extrapolation error in the learned reward model. Because the reward model is trained only on a small set of preference-labeled trajectories, it must generalize to a much larger, unlabeled offline dataset containing many out-of-distribution (OOD) state-action pairs. In these OOD regions, even small modeling errors can be arbitrarily amplified during policy optimization, leading to unstable and poorly performing policies. Existing methods—PT, MR, IPL, HPL,CPL,and DTR—either (i) attempt to fit a more accurate global reward model or (ii) avoid explicit reward models entirely, but crucially, none provides a mechanism to actively constrain the policy to preference-supported regions of the state-action space.
>
> SPOT directly targets this extrapolation problem through a novel approach. We leverage PT's attention mechanism to extract preference-aligned subgoals (high-attention, high-reward states where human preferences are most reliable). We then train a state-action-conditioned CVAE that, given any offline state-action pair, predicts a distribution over these subgoals. This enables us to define a subgoal-conditioned shaping term that locally increases reward only when trajectories remain close to preference-supported subgoal regions.
>
> The CVAE is not simply a modular addition—it is an architecturally necessary component for this approach. During policy learning, we need to utilize arbitrary state-action pairs sampled from the offline dataset to train agent. Simply extracting attention weights from PT or relying on the offline dataset alone cannot address extrapolation error, as there is no mechanism to connect unlabeled offline samples to preference-supported regions identified by human feedback. **The CVAE solves this fundamental problem by providing a learnable mapping from any offline state-action pair to preference-supported subgoal distributions.** By conditioning on state-action pairs and learning to predict distributions over attention-derived subgoals, the CVAE effectively learns **"where human preference is reliable"** for any point in the offline dataset. This enables us to define a subgoal-conditioned shaping term that locally increases reward only when trajectories remain close to preference-supported subgoal regions—directly preventing extrapolation.
>
> To our knowledge, no prior offline PbRL method uses attention-derived subgoals combined with generative mapping to drive reward shaping as a targeted countermeasure to extrapolation error. This represents a conceptually distinct contribution: rather than improving reward model accuracy globally or avoiding rewards altogether, we actively guide policies to regions where preferences are demonstrably reliable. **Overall, this framework represents a novel, simple yet effective approach to systematically mitigate extrapolation error in offline preference-based RL—a problem that has remained fundamentally unsolved in prior work.**
>
> ---
>
> **Computational Overhead**
>
>
> Regarding computational cost,as detailed in Appendix C.2, the subgoal CVAE adds only small amount of time compared to PT baseline, while methods like HPL incur 7–10× longer training times. The extra computation is small relative to the substantial gains in both stability and final performance that SPOT achieves across all benchmarks. Further details about computational overhead is described as below in Q3.
>
> We hope this clarifies both the novelty of our approach and the practical efficiency of the method. We would be happy to further discuss any remaining concerns.

---

> ### Author Response · Authors · 2025-11-25
>
> # Q2: Further experimental settings and baselines
> Thank you for this important feedback. We have carefully addressed your concerns regarding experimental setting and baseline comparisons.
>
> Regarding AntMaze experiments, we acknowledge that PT included tasks with original sparse rewards such as AntMaze. However, previous work (e.g., DPPO [1]) has reported **a critical bug in the D4RL AntMaze benchmark** that affects the reliability of reported scores. Following this discovery, most subsequent offline RL and PbRL papers—including recent methods like IPL, HPL,CPL,and DTR—have stopped reporting AntMaze results. We therefore followed this established practice and instead focused on MuJoCo locomotion and manipulation tasks (Robosuite, Meta-World).
>
> Following your suggestion, we have added CPL [2] and DTR [3] to our comparisons and updated the baseline results throughout the paper. The updated results in Table 1 show that SPOT achieves the best average normalized return (78.82) across all 10 tasks, outperforming all baselines including the recent CPL (44.98) and DTR (54.08) methods. **SPOT demonstrates competitive or superior performance on individual tasks, with particularly strong improvements on challenging manipulation tasks**. Additionally, SPOT exhibits lower variance (7.76 avg std) compared to most baselines, indicating more stable learning—a key advantage when mitigating extrapolation error.
>
> We believe these additional baseline experiments substantially strengthen our method empirically  and would be happy to discuss any specific additional baselines or tasks you feel would further improve the evaluation.
>
> | Dataset                      | Oracle        | MR            | PT            | IPL           | HPL           | CPL           | DTR           | SPOT (ours)   |
> |------------------------------|---------------|---------------|---------------|---------------|---------------|---------------|---------------|---------------|
> | D4RL Locomotion Tasks        |               |               |               |               |               |               |               |               |
> | hop-m-r                      | 92.02 ± 7.23  | 37.21 ± 12.53 | 52.15 ± 25.94 | 74.96 ± 5.79  | 79.89 ± 10.01 | 62.21 ± 6.40  | **94.18 ± 0.28**  | 85.08 ± 1.32  |
> | hop-m-e                      | 62.10 ± 30.42 | 63.60 ± 25.42 | 74.46 ± 4.33  | 42.11 ± 8.93  | 95.30 ± 10.66 | 44.97 ± 44.74 | **102.12 ± 6.79** | **98.73 ± 7.50**  |
> | walk-m-r                     | 67.59 ± 7.91  | 71.39 ± 2.66  | 73.85 ± 3.18  | 47.05 ± 15.24 | 49.89 ± 10.49 | 36.10 ± 12.61 | 69.09 ± 4.85  | **76.89 ± 2.46**  |
> | walk-m-e                     | 108.72 ± 1.86 | **110.88 ± 0.76** | **110.6 ± 0.43**  | 107.78 ± 0.95 | 103.14 ± 2.49 | 108.98 ± 0.15 | **110.96 ± 0.37** | **110.06 ± 0.28** |
> | Robosuite Manipulation Tasks |               |               |               |               |               |               |               |               |
> | lift-mh                      | 81.62 ± 5.54  | **95.62 ± 2.23**  | 68.46 ± 10.02 | 84.49 ± 4.28  | **88.37 ± 3.06**  | 18.79 ± 5.19  | 22.30 ± 21.96 | 65.17 ± 12.57 |
> | lift-ph                      | **98.43 ± 1.15**  | 87.40 ± 10.65 | **95.50 ± 1.90**  | **95.81 ± 3.04**  | 61.04 ± 7.61  | 28.41 ± 5.85  | 9.86 ± 4.31   | **97.12 ± 1.81**  |
> | can-mh                       | 34.30 ± 6.95  | 47.95 ± 2.29  | 53.06 ± 14.48 | 41.12 ± 2.21  | 35.19 ± 12.25 | 12.34 ± 5.44  | **60.28 ± 2.56**  | **60.55 ± 1.65**  |
> | can-ph                       | **73.25 ± 2.70**  | 51.90 ± 6.58  | 48.74 ± 5.82  | **67.98 ± 3.41**  | 10.90 ± 4.33  | 9.15 ± 2.40   | 39.82 ± 8.25  | **63.82 ± 5.64**  |
> | Meta-World Manipulation Tasks|               |               |               |               |               |               |               |               |
> | drawer-open                  | —             | 86.6 ± 14.3   | 42.8 ± 29.1   | **87.64 ± 6.99**  | 83.13 ± 12.64 | 75.48 ± 7.42  | 26.90 ± 24.09 | 66.80 ± 18.05 |
> | plate-slide                  | —             | 51.5 ± 11.9   | 51.0 ± 2.8    | 51.18 ± 6.63  | 28.73 ± 12.22 | 53.41 ± 4.94  | 5.24 ± 5.07   | **64.0 ± 4.1**    |
> | Average                      | 77.25         | 73.61         | 74.76         | 73.24         | 67.96         | 44.98         | 54.08         | **78.82**         |
> | Avg. Std                     | 11.89         | 11.51         | 13.80         | **6.95**          | 9.36          | 9.51          | **7.85**          | **7.76**          |
> ---
> [1]: AN, Gaon, et al. Direct preference-based policy optimization without reward modeling. Advances in Neural Information Processing Systems, 2023.
>
> [2] : Contrastive Preference Learning: Learning from Human Feedback without Reinforcement Learning. 2024.
>
> [3] : In-Dataset Trajectory Return Regularization for Offline Preference-based Reinforcement Learning. AAAI 2025.

---

> ### Author Response · Authors · 2025-11-25
>
> # Q3 : Details on computational overhead
>
> Thank you for raising this important practical concern. We already included detailed training time measurements in Appendix C.2. While we addressed the computational aspects in our response to Q1, we provide additional clarification on the computational overhead here.
>
> As reported in the Appendix C.2 and summarized in the table below, CVAE introduces only a slight computational overhead . On representative tasks (hopper-m-e, walker2d-m-r), the total time of SPOT is approximately 10–15% larger than PT (e.g., hopper-m-e: 5,196s → 5,824s; walker2d-m-r: 10,037s → 11,711s).
>
> **We measured training times and found that SPOT introduces negligible computational overhead compared to existing methods,  achieving superior performance.**
> These measurements demonstrate that adding our subgoal CVAE and shaping term does not fundamentally change the computational regime of offline PbRL. The overhead is far smaller than the substantial cost you were concerned about, while the performance gains shown in Table 1 remain significant. We believe that 10–15% increase in training time is a reasonable trade-off for the improved sample efficiency, stability, and final performance that SPOT provides across all benchmark tasks.
>
> | Method         | Environment  | Reward Model Training | Offline RL Training | Total Time        |
> |----------------|--------------|----------------------|---------------------|-------------------|
> | IPL            | hopper-m-e   | --                   | 4:19:07 (15,547s)   | 4:19:07 (15,547s) |
> | HPL            | hopper-m-e   | 4:30:37 (16,237s)    | 5:47:28 (20,848s)   | 10:18:05 (37,085s)|
> | CPL            | hopper-m-e   | 0:19:03 (1,143s)     | 1:14:21 (4,462s)    | 1:33:24 (5,605s)  |
> | PT (baseline)  | hopper-m-e   | 0:22:54 (1,374s)     | 1:03:42 (3,822s)    | 1:26:36 (5,196s)  |
> | SPOT (ours)    | hopper-m-e   | 0:31:18 (1,878s)     | 1:05:45 (3,946s)    | 1:37:03 (5,824s)  |
> |              |              |                      |                     |                   |
> | IPL            | walker-m-r   | --                   | 4:03:06 (14,586s)   | 4:03:06 (14,586s) |
> | HPL            | walker-m-r   | 4:40:35 (16,835s)    | 5:49:07 (20,947s)   | 10:29:42 (37,782s)|
> | CPL            | walker-m-r   | 1:20:47 (4,847s)     | 1:18:36 (4,715s)    | 2:39:23 (9,562s)  |
> | PT (baseline)  | walker-m-r   | 1:44:17 (6,257s)     | 1:02:59 (3,780s)    | 2:47:16 (10,037s) |
> | SPOT (ours)    | walker-m-r   | 2:07:14 (7,634s)     | 1:07:57 (4,077s)    | 3:15:11 (11,711s) |

---

### Author Response · Authors · 2025-12-04

We sincerely thank the reviewers for their thoughtful feedback. We revised the paper to clarify the core contribution, strengthen experiments, and improve clarity. Below we summarize the key points.

---

### 1. Conceptual Contribution & Novelty (Extrapolation Error)

- We explicitly position SPOT as a method for **mitigating extrapolation error** in offline PbRL: existing methods either fit a global reward model (PT, MR, HPL, DTR) or avoid explicit rewards(IPL,CPL), but **none explicitly keep the policy within preference-supported regions** of the state–action space to avoid extrapolation error.
- The **state–action–conditioned CVAE** is essential: it maps any offline state-action $(s,a)$ to a distribution over subgoals, enabling a **subgoal-shaped reward**

    $r_{\text{final}} = r_{\text{model}} + \lambda r_{\text{shape}}$

    that regularizes OOD exploitation while preserving the reward model’s generalization.


---

### 2. Experimental Coverage & Added Baselines (CPL, DTR)

- We **added CPL and DTR** to the main table.  SPOT still demonstrates the best average performance over 10 tasks (78.82)—substantially outperforming PT (74.76), IPL(73.24),DTR (54.08), and CPL (44.98)— while also showing lower average standard deviation, indicating more stable learning.
- Following prior work citing a D4RL AntMaze bug, we omit AntMaze and instead rely on **Robomimic and Meta-World**, which are **sparse-reward, goal-based** benchmarks (binary success) and thus appropriate for testing subgoal guidance.

---

### 3. Computational Overhead

- We report detailed timings (Appendix C.2): SPOT incurs only negligible extra training time over PT on representative tasks.
- For comparison, **HPL requires 7–10×** longer training, so the additional CVAE introduces only **minor overhead** relative to the performance and stability gains.

---

### 4. Attention Reliability, Limited Feedback, and Noise Assumptions

- SPOT **does not rely on raw attention alone**: subgoals must satisfy a **dual filter** (top-K% attention **and** above-average reward with temporal structure), which removes spurious high-attention states.
- In section 5.5, under **limited feedback**, we show that SPOT remains strong performance while PT degrades. We follow the **standard offline PbRL assumption of clean preferences** and mark noisy-label extensions as future work.

---

### 5. Subgoal Extraction, Top-K Selection, and Meaningfulness

- In section 5.2.1, we already report that top 10% attention states clearly outperform lower percentiles in both mean return and variance, empirically justifying K = 10\%
- We extend  **Can-ph visualizations** and previously mentioned **latent-space plots** (t-SNE, PCA, UMAP). Subgoals show a consistent **one-step-ahead** structure and stay close to the observation manifold (precision ≈ 0.65, recall ≈ 0.80), indicating meaningful, task-relevant waypoints.

---

### 6. Cosine Similarity: Theory and Objective Correction

- We ground the cosine similarity term in **distance-based reward shaping** (e.g., ROLeR), interpreting it as enforcing semantic proximity between generated and target subgoals.
- We correct the **sign** in the objective so that similarity is properly maximized, and ablations show that this term improves performance over reconstruction-only CVAE and alternative shaping choices.

---

### 7. Algorithmic Clarity: Eq. (5), Subgoal Set, and Pseudocode

- We clarify that Eq. (5) defines a **deterministic subgoal set** (no random sampling) based on attention and reward thresholds; all subgoals are used to build triplets $(s_t, a_t, g_t)$ between consecutive waypoints.
- We add **full pseudocode (Algorithm 1)** covering subgoal extraction, CVAE training, and offline RL with subgoal-shaped rewards, making the pipeline easy to follow.

---

### 8. Manipulation Metrics and IL vs RLHF (Diffusion Policy)

- We clarify that Diffusion Policy is a **pure imitation learning** method that can reach near-100% success on expert data, while SPOT is designed for the **offline preference-based RL (RLHF)** setting with learned rewards. Our benchmarks follow the **standard offline PbRL testbed**, and our focus is improving performance within this RLHF formulation rather than replacing IL where imitation is already near-optimal.

---

We appreciate the reviewers’ feedback, which helped us sharpen the conceptual story, strengthen comparisons , and improve the clarity of SPOT’s design and evaluation.

---

### Meta-Review · Area_Chair_cEuK · 2026-01-04

**Summary:**

All the original reviews are not positive and they raised many issues related to significance, novelty and evaluation. The author response (which is quite detailed) addressed some of concerns. But overall, the quality of the paper seems slightly below the bar of ICLR.

**Reviewer Concerns:**

Some concerns are not fully addressed such as the robustness of the attention mechanism and the benchmark’s necessity of the RLHF paradigm over Imitation Learning (IL).

**Reviewer Scores:**

Most reviewers will slightly increase their scores.

---

### Decision · Program_Chairs · 2026-01-26

Reject